# Borderline Oxacillin-Resistant *Staphylococcus aureus* (BORSA) Bacteremia—Case Report

**DOI:** 10.3390/antibiotics14080809

**Published:** 2025-08-07

**Authors:** Beverly Buffart, Philippe Clevenbergh, Alina Stiuliuc, Ioannis Raftakis, Mony Hing, Véronique Yvette Miendje Deyi, Olivier Denis, Delphine Martiny, Nicolas Yin

**Affiliations:** 1Department of Microbiology, Laboratoire Hospitalier Universitaire de Bruxelles—Universitair Laboratorium Brussel (LHUB-ULB), Université Libre de Bruxelles (ULB), 1000 Brussels, Belgium; 2Infectious Diseases Department, University Hospital Brugmann, Université Libre de Bruxelles (ULB), 1090 Brussels, Belgium; 3Rhumatologic Diseases Department, University Hospital Brugmann, Université Libre de Bruxelles (ULB), 1090 Brussels, Belgium; 4Faculty of Medicine, Université libre de Bruxelles (ULB), 1070 Brussels, Belgium; 5Faculty of Medicine and Pharmacy, University of Mons (UMONS), 7000 Mons, Belgium; 6National Reference Centre for *Staphylococcus aureus*, Laboratoire Hospitalier Universitaire de Bruxelles—Universitair Laboratorium Brussel (LHUB-ULB), Université Libre de Bruxelles, 1000 Brussels, Belgium

**Keywords:** *Staphylococcus aureus*, methicillin resistance, bacteremia, BORSA

## Abstract

Introduction: Borderline oxacillin-resistant *Staphylococcus aureus* (BORSA) represents a rare and poorly characterized phenotype of *S*. *aureus*. Its detection remains challenging, even in modern clinical laboratories. Moreover, there is no consensus on the optimal therapeutic approach, and treatment strategies remain controversial. In this report, we present a rare case of BORSA bacteremia and discuss potential approaches to improve its detection and management. Case presentation: A 39-year-old woman with systemic lupus erythematosus was admitted for a suspected exacerbation, complicated by multiple serositis and nephritis. She was on chronic treatment with methylprednisolone and hydroxychloroquine. On admission, she was afebrile. Laboratory investigations revealed elevated C-reactive protein and increased D-dimer levels. Later, she developed a septic peripheral venous thrombophlebitis, and treatment was adjusted to amoxicillin–clavulanate. Blood cultures grew *S. aureus*, prompting a switch to intravenous oxacillin based on a negative penicillin-binding protein 2a test. A discrepancy in the antimicrobial susceptibility test was observed, with cefoxitin showing susceptibility and oxacillin resistance. Further characterizations were carried out, confirming a BORSA infection. Treatment was switched to linezolid and ciprofloxacin with good recovery. Conclusions: This case highlights the complexity of managing a patient with an uncommon and poorly documented infection. The lack of data on BORSA infections and the difficulties in detecting and treating them led to a prolonged delay in the appropriate management of this patient.

## 1. Introduction

*Staphylococcus aureus* naturally colonizes the nasal cavities, skin, throat, and other mucous membranes of humans and warm-blooded animals. It is responsible for a wide range of infections, from mild tissue infections to severe invasive diseases such as bacteremia, endocarditis, pneumonia, and osteomyelitis, which are associated with significant morbidity and mortality [1]. Treatment strategies for these infections vary according to the susceptibility of the strain to oxacillin. The well-known methicillin-resistant *S. aureus* (MRSA) remains a major public health concern. Resistance is typically mediated by the acquisition of the staphylococcal chromosomal cassette, SCC*mec*, which carries the *mecA* or *mecC* gene. These genes encode an altered penicillin-binding protein (PbP2a) with reduced affinity for beta-lactams, allowing cell wall synthesis to continue even in the presence of otherwise inhibitory antibiotic concentrations [2].

In contrast, a much less common and less understood phenotype of *S. aureus* is borderline oxacillin-resistant *S. aureus* (BORSA). BORSA strains exhibit reduced susceptibility to oxacillin with minimal inhibitory concentrations (MIC) ranging from 1 to 8 mg/L. While *S. aureus* strains with MICs > 2 mg/L are typically considered methicillin-resistant due to the presence of *mecA* or *mecC*, BORSA strains do not carry these genes. The mechanism of resistance in BORSA remains under investigation and may involve either overproduction of beta-lactamase [3] or point mutations in the native PBP genes [4]. Currently, the European Committee on Antimicrobial Susceptibility Testing (EUCAST) does not recommend routine screening for BORSA [5] although such strains are often clinically assimilated to MRSA [6]. This provides an explanation for the probable underestimation of BORSA cases. Moreover, detection remains challenging due to absence of specific, standardized identification methods. Some authors suggest that BORSA infections are generally more severe than those associated with Methicillin-sensitive *S. aureus* (MSSA) are, although this remains a topic of ongoing debate in the scientific community [6]. There are currently no clear official guidelines regarding the treatment of BORSA infections. Some authors have hypothesized that BORSA could be effectively treated with penicillinase-resistant penicillins (PRPs) such as cloxacillin and that the BORSA phenotype does not correlate with in vivo resistance [7]. However, others have reported treatment failures with high-dose cloxacillin, advocating instead for alternative agents such as vancomycin [8]. Here, we describe a case of BORSA bacteremia in a 39-year-old woman. We also analyze the genetic characteristics of the isolate and perform a retrospective analysis of BORSA strains submitted to the Belgian National Reference Centre for *S. aureus*.

## 2. Case Presentation and Results

A 39-year-old woman was admitted for a suspected exacerbation of systemic lupus erythematosus, diagnosed one year earlier. The case was complicated by multiple serositis and nephritis. Her chronic treatment included methylprednisolone (32 mg daily) and hydroxychloroquine (200 mg twice daily). On examination, the patient was afebrile but complained of chest pain increased by breathing. Laboratory results showed 5690 leucocytes/µL, neutrophils 79%, C-reactive protein (CRP) 19 mg/dL (reference value < 5 mg/dL), estimated glomerular filtration rate (CKD-EPI) > 90 mL/min, normal hepatic enzymes, and increased D-Dimer levels at 4318 ng/mL (reference value < 500). A chest CT scan showed no pulmonary embolism, bilateral pleural effusion with passive atelectasis and hypoventilation. No blood culture were collected at admission. Nasal screening for MRSA carriage was negative. Bronchoalveolar lavage cultures were negative for common bacteria and *Mycobacterium tuberculosis*, and respiratory multiplex Polymerase Chain Reaction (PCR) panel (Taqman™ Array Card) did not detect atypical bacteria, viruses or fungi [9]. One week later, she developed peripheral venous septic thrombophlebitis following catheter placement. Intravenous amoxicillin–clavulanate 1 g, 4 four times daily was initiated. Two sets of blood cultures were collected on hospital day (HD) 7 and incubated using the BD Bactec™ FX system (BD diagnostics, Franklin Lakes, NJ, USA). After 17 h of incubation (HD 8), both sets grew Gram-positive cocci in clusters, later identified as *S. aureus* using the MALDI Biotyper^®^ sirius IVD system (version 4.1.100, Bruker Daltonics, Bremen, Germany). The Clearview™ PBP2a SA Culture Colony Test (Abbott Diagnostics, Scarborough, ME, USA) was negative. Consequently, treatment was switched to intravenous oxacillin 2 g four times daily. However, antimicrobial susceptibility testing performed with the Vitek^®^ 2 system, (Biomérieux, Marcy-L’étoile, France) using EUCAST guidelines 2023 (version 13.0) showed a discrepancy: the cefoxitin screen was negative while the oxacillin Vitek^®^ 2-derived MIC value was ≥4 mg/L. Therefore, the expert system interpreted the strain as resistant to oxacillin. The isolate was also resistant to cotrimoxazole and tetracycline, susceptible to linezolid and susceptible (increased exposure) to ciprofloxacin. (Table 1) Due to difficult intravenous access, antibiotic therapy was transitioned on HD 9 to oral linezolid (600 mg twice daily) and ciprofloxacin (750 mg twice daily). Treatment was continued for 14 days from the first negative blood culture (Figure 1). Transthoracic cardiac echography was normal.

Further characterization of the isolate was performed within the framework of the Belgian National Reference Center (NRC) for *S. aureus* (LHUB-ULB, Brussels, Belgium). Oxacillin and cefoxitin MICs were both 4 mg/L, as determined with the E-test^®^ (Biomérieux, Marcy-L’étoile, France) on Mueller–Hinton agar (supplemented with 2% NaCl for oxacillin MIC) [10] and Sensititre™ (Thermo Fisher, Cleveland, OH, USA) performed according to the manufacturers’ instructions. Disk diffusion with a 30 µg cefoxitin disk yielded an inhibition zone of 27 mm. *mecA* and *mecC* were not detected with end-point PCR of the strain using previously described methods [11,12,13]. To further investigate the genetic basis of resistance, whole-genome sequencing was performed. Genomic DNA was extracted using EZ1 & 2 Virus Mini Kit v2.0 (Qiagen, Hilden, Germany) and the EZ2 Connect MDx instrument (Qiagen). DNA was enzymatically fragmented and modified to generate an Illumina compatible DNA library using Revelo DNA-Seq for MagicPrep NGS (Tecan, Männedorf, Switzerland). The library was sequenced using a MiniSeq machine (Illumina Inc., San Diego, CA, USA) with MiniSeq Mid Output Kit (300 cycles) in 2 × 150 base pairs (bp) paired mode. De novo genome assembly was carried out using the Velvet algorithm and core genome multilocus sequence type was determined using Ridom SeqSphere+ version 10.0.5 (Ridom GmbH, Münster, Germany). The assembly was screened for acquired antimicrobial resistance genes and chromosomal point mutations using ResFinder version 4.6.0 [14]. The isolate belonged to the sequence type ST1 and the cgMLST complex type CT42566. It carried the resistance genes *tet*(K) and *dfr*G and exhibited multiple point mutations in the genes encoding PBP2 (E315A, A576S and A606D) and PBP4 (T25A, T189S, L234H, T409A). The isolate did not carry *bla*Z or *mec* genes. The genome assembly was deposited at the National Center for Biotechnology Information (NCBI) under BioSample accession number SAMN47436409.

## 3. Discussion

BORSA remains a relatively rare entity, with reported prevalence in the literature ranging from 1.4% to 12.5% [6]. A literature review was conducted to examine previously published cases of BORSA bacteremia and/or endocarditis. To ensure the reliability of the data, studies that did not assess the presence or absence of *mecA* gene were excluded. To date, only a few cases of bacteremia have been documented. Notably, the majority were associated with a cutaneous portal of entry (Table 2). In our case, the patient also presented a cutaneous portal of entry, cellulitis of the hand, following peripheral venous catheter insertion. While this observation may be coincidental, it is noteworthy that several reported BORSA cases and outbreaks have been linked to dermatology units or patients with dermatological conditions [15,16,17,18].

To date, the detection of BORSA remains complex and insufficiently investigated. These strains may be misclassified as susceptible to oxacillin based on the inhibition zone produced by the cefoxitin disk. Simultaneous antimicrobial susceptibility testing for both cefoxitin and oxacillin may be suggested to ensure that no BORSA strain is missed [6]. It can be challenging to differentiate between MRSA and BORSA when the oxacillin MIC is ≥8 µg/L. In such cases, additional testing is warranted, including the detection of PBP2a via latex agglutination and/or molecular detection of the *mecA* and *mecC* genes by PCR. At the Belgian NRC, BORSA detection follows a two-step strategy: initial MIC determination for oxacillin and cefoxitin using the E-test^®^, followed by the detection of *mecA* and *mecC*. Isolates resistant to oxacillin and/or cefoxitin but negative for *mec* genes are classified as BORSA [21]. Buchan et al. demonstrated that BORSA strains could also be detected using a specific chromogenic medium initially developed for MRSA detection [22]. However, implementing such media may be logistically complex and costly. Therefore, we investigated whether BORSA strains could be detected using only the diagnostic tools currently available in our laboratory. To this end, the Belgian NRC provided four confirmed BORSA isolates. The Belgian NRC for *S. aureus* is a center that voluntarily analyses and characterizes unusual phenotypes submitted by clinical laboratories across Belgium. As a significant number of laboratories routinely dispatch strains for investigation, these facilities have accumulated a substantial collection of strains displaying atypical phenotypes, including BORSA. We retested the four NRC-confirmed BORSA isolates (NRC-4 is the strain from this case) using the Vitek^®^ 2 system in our laboratory. All strains exhibited a discrepancy between cefoxitin screen results and oxacillin MICs (Table 3). Despite this, the Vitek^®^ 2 system classified them as MRSA. The same observation was made for all BORSA strains associated with endocarditis and/or bacteremia reported in the literature that were analyzed with Vitek-2 and for which a cefoxitin screen was available (Table 2).

The latest articles [20,22] investigating the use of VITEK-2 for antibiotic susceptibility testing describe also a consistent discrepancy. Specifically, *S. aureus* strains were flagged as BORSA if the cefoxitin screen was negative, yet the strain was identified as methicillin-resistant based on an MIC of 4 mg/L or higher. Unlike whole-genome sequencing (WGS) and PCR, this would be an inexpensive alternative that could be easily applied routinely.

Whole-genome sequencing of the strain revealed the presence of *tet*(K) and *dfr*G, responsible for phenotypic resistance to tetracycline and trimethoprim, respectively. Multiple point mutations were identified in *pbp2* (E315A, A576S, and A606D) and *pbp4* (T25A, T189S, L234H, and T409A). Whilst resistance in BORSA is sometimes attributed to an overexpression of Beta-lactamases, this is not the case in our patient given the absence of the *blaZ* gene. Several of the mutations identified in our case strain (A606D, T25A, L234H, T409A, and E315A) have previously been reported in WGS studies of *S. aureus* strains [23,24,25]. However, in all these cases, they were associated with MRSA, making our results difficult to interpret. It could be that all these genes are indeed associated with a reduced susceptibility to oxacillin, which was masked by the presence of *mecA*. Studies investigating *S. aureus* strains lacking *mecA* and *blaZ* in which pbp gene sequencing was performed are scarce. One such study by Hackbarth et al. first described the A576S mutation in 1995. However, this mutation was not associated with increased penicillin resistance, as MIC values for penicillin remained unchanged (0.04 mg/L) in strains differing only by the presence of A576S [26]. Our findings do not allow definitive conclusions regarding the role of individual mutations in the acquisition of oxacillin resistance, with the exception of A576S, which appears to have no such effect. Further studies on BORSA strains would help to elucidate this.

Official treatment guidelines for *S. aureus* bacteremia are stratified based on methicillin susceptibility. For MSSA, first-line therapy consists of a penicillinase-resistant beta-lactam (e.g., oxacillin, flucloxacillin, or cefazolin), while MRSA infections are typically managed with vancomycin (or possibly daptomycin). Treatment duration depends on clinical context but generally lasts 14 days for uncomplicated *S. aureus* bacteremia [27].

The diversity and complexity of the therapeutic approaches reported for BORSA bacteremia and endocarditis (Table 2) clearly reflect a lack of consensus for the management of BORSA infections. While some studies still suggest that beta-lactams may still be effective, others report treatment failures, necessitating the use of alternative agents. In the absence of specific treatment recommendations for BORSA bacteremia, therapeutic decisions are made on a case-by-case basis, often involving different antimicrobials and yielding variable outcomes.

## 4. Conclusions

This case highlights the diagnostic and therapeutic challenges posed by BORSA, which is often misclassified by automated systems. Our investigation indicates that discrepancies between oxacillin and cefoxitin susceptibility may serve as a potential warning sign and should prompt further testing for BORSA. These findings underscore the need to integrate both phenotypic and molecular approaches for accurate detection and appropriate clinical management. Therapeutic strategies remain inconsistent, as standard treatment guidelines are lacking. Our findings support individualized management and call for further genomic and clinical research to define BORSA’s resistance mechanisms and inform standardized diagnostic and therapeutic protocols.

## Figures and Tables

**Figure 1 antibiotics-14-00809-f001:**
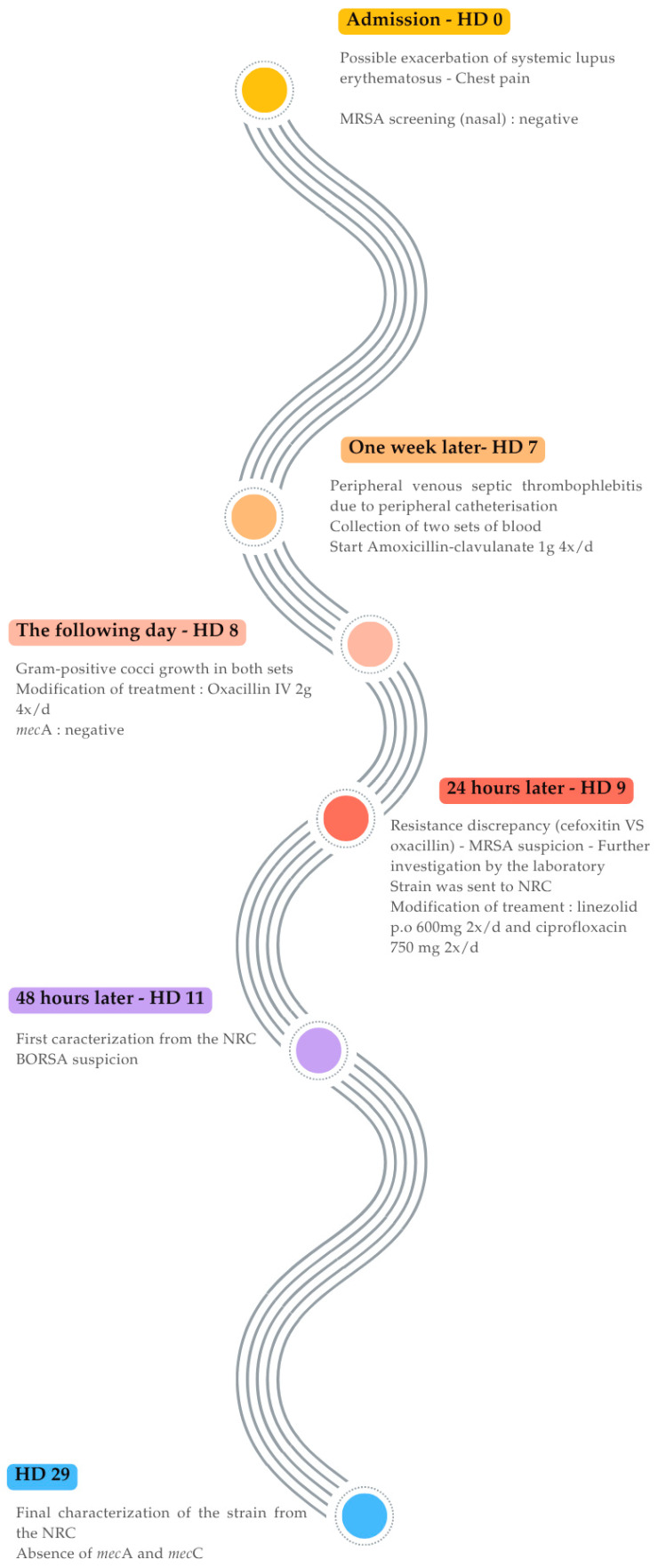
Timeline of therapeutic, microbiological, and treatment events in patient care.

**Table 1 antibiotics-14-00809-t001:** Antibiogram of our stain of *S. aureus* supplied by Vitek-2 using EUCAST 2023 guidelines.

Antibiotics	Susceptible (S)/Susceptible, Increased Exposure (I)/Resistant (R)	Vitek-2^®^-Derived MIC Values (mg/L)
Oxacillin	S	≥4
Ciprofloxacin	I	≤0.5
Trimethoprim/Sulfamethoxazole	R	160
Tobramycin	S	≤1
Gentamycin	S	≤0.5
Erythromycin	S	0.5
Clindamycin	S	0.25
Minocycline	S	≤0.5
Tetracycline	R	≥16
Linezolid	S	2
Rifampicin	S	≤0.03
Fusidic acid	S	≤0.5
Vancomycin	S	1
Kanamycin	S	≤4
Teicoplanin	S	≤0.5
Mupirocin	S	≤1

**Table 2 antibiotics-14-00809-t002:** Clinical and microbiological characteristics of various BORSA strains responsible for bacteremia and/or endocarditis referenced in the literature.

	Clinical Sample	Diagnosis–Origin of Infection–Other Pathological Condition	Cefoxitin Screen (mm)	Oxacillin MIC (mg/L)	Treatment	Outcome	References
1	Blood	Endocarditis–involvement of prosthetic material	/	≥4 ^a^ 2–4 ^b^	High-dose flucloxacillin and then vancomycin	Death	[15]
2	Blood	Bacteremia and possible endocarditis–infected venous line–kidney transplantation	/	≥4 ^a^ 4 ^b^	Unknown	Unknown	[15]
3	Blood	Bacteremia and infective endocarditis complicated with septic arthritis and pneumonia	27 (S) ^e^	≥4 ^a^	IV vancomycin 1 g B.I.D. and oral TMP-SMX 1440 mg BD for 42 days	Recovery	[19]
4	Synovial		29 (S) ^e^	≥4 ^a^			[19]
5	Blood	Abscess and bacteremia	27 (S) ^e^	≥4 ^a^	IV cloxacillin 2 g QID for 1 day; then, IV vancomycin 1 g BD for 13 days	Recovery	[19]
6	Pus		26 (S) ^e^	≥4 ^a^			[19]
7	Blood	Pneumonia and bacteremia	27 (S) ^e^	≥4 ^a^	IV vancomycin 1 g loading dose, 750 mg/day (renal adjusted dosing) for 5 days, then IV ceftaroline 300 mg TDS for 11 days	Recovery	[19]
8	Blood	Catheter-related bloodstream infection and bacteremia	28 (S) ^e^	≥4 ^a^	IV vancomycin 1 g loading dose; then, 750 mg/day for 14 days	Recovery	[19]
9	Blood	Bacteremia and infective endocarditis	27 (S) ^e^	≥4 ^a^	IV vancomycin 1 g BD and IV metronidazole 500 mg TDS for 4 days	Unresolved infection—patient requested to be discharged	[19]
10	Blood	Bacteremia, infective endocarditis, and vertebral osteomyelitis–invasive material: intravenous drug user, tricuspid valve replacement (bioprosthetic valve), and pacemaker insertion	/	≥4 ^a^ 12 ^c^	IV cloxacillin 2 g every 4 h and 600 mg of rifampin orally once daily and then vancomycin	Symptoms resolved—patient requested to be discharged	[8]
11	Blood	Community-acquired BORSA bacteremia; infective endocarditis and lung abscesses–chronic eczema; cellulitis in the left leg	/	4 ^b^	IV cloxacillin (2 g every 6 h) was given on days 2–5, and then IV vancomycin + rifampicin on day 5. Treatment switched to ampicillin/sulbactam (3 g every 6 h) on day 10 (and for 6 weeks) with rifampin; vancomycin treatment was stopped	Condition progressively deteriorated from day 2 to day 10 and defervescence occurred 3 days later	[7]
12a	Blood	Sternal wound abscess, bacteremia, and infective endocarditis–bioprosthetic aortic valve replacement?	≈S ^a^	≥4 ^a^, and then the MIC was suppressed	Empirical vancomycin (15 mg/kg intravenously, every 24 h [i.v. q24h] at a separate outside institution. Two days later, vancomycin was de-escalated to cefazolin (2 g i.v. q8h). A diagnosis of IE was made: the treatment switched to oxacillin (2 g i.v. q4h) and synergistic gentamicin (1 mg/kg i.v. q12h), with plans to add rifampin. Patient was transitioned from oxacillin to cefazolin due to rising serum creatinine on HD9, then transitioned to daptomycin (8 mg/kg i.v. q24h) and rifampin (300 mg orally [p.o.] q8h) on HD11 and for 6 weeks, and finally, to lifelong suppressive doxycycline (100 mg p.o. q12h)	Recovery	[20]
12b			≈S ^a^	“S” ^a^ (MIC was suppressed)			[20]
12c			21 (R) ^e^	≥2 ^b^			[20]
13	Blood	Unknown	33 (S) ^e^	2 ^d^	Not specified for each individual case—treatments were pristinamycin, cefotaxime, or imipenem	Recovery	[17]
14	Blood	Unknown	28 (S) ^e^	4 ^d^			[17]
15	Blood	Unknown	32 (S) ^e^	2 ^d^			[17]
16	Blood	Unknown	30 (S) ^e^	2 ^d^			[17]
17	Blood	Bacteremia–Dermatitis atopica	/	“Reduced” (disk diffusion) ^e^	Dicloxacillin	Recovery	[18]
18	Blood	Bacteremia–Mycosis fungoides	/	“Reduced” (disk diffusion) ^e^			[18]
19	Blood	Bacteremia–Mb. Darier	/	“Reduced” (disk diffusion) ^e^			[18]
20	Blood	Bacteremia–Pemphigoides bullosa	/	“Reduced” (disk diffusion) ^e^			[18]

^a^ Vitek2-derived MIC (mg/L); ^b^ determined with an E-test on Mueller–Hinton agar with 2% NaCl (mg/L); ^c^ determined with an E-test (without specifying whether NaCl was added to the MH); ^d^ determined via broth microdilution without the use of Vitek-2; ^e^ determined via disk diffusion (mm): cefoxitin 30 µg disk diffusion and oxacillin 1 µg disk diffusion; B.I.D.: bis in die; (S): susceptible; (R): resistant.

**Table 3 antibiotics-14-00809-t003:** Microbiological characteristics of BORSA strains from the National Reference Center.

Isolates	Cefoxitin Screen	Oxacillin MIC E-test (mg/L)	Vitek-2^®^-Derived Oxacillin MIC Vitek2 (mg/L)	*mec* Gene Investigation (*mec*A and *mec*C)
NRC-1	26 mm	4	≥4	Absence
NRC-2	26 mm	4	≥4	Absence
NRC-3	23 mm	4	≥4	Absence
NRC-4	27 mm	4	≥4	Absence

## Data Availability

All data generated or analyzed during this study are included in this published article.

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
