# Peer review of "Borderline Oxacillin-Resistant Staphylococcus aureus (BORSA) Bacteremia—Case Report"

_antibiotics, 2025, doi:10.3390/antibiotics14080809_

Round 1
Reviewer 1 Report
Comments and Suggestions for Authors
This manuscript describes a case of BORSA. in a 39 old woman, analyse the genetic characteristics of the incriminated isolate and perform a retrospective analysis of BORSA strains submitted to the Belgian National Reference Centre for S. aureus. The manuscript is relevant, interesting, and addresses BORSA infections and their alternative therapeutic management. However, the manuscript requires improvement before consider for publication.
Comment for authors
- Line 2: pleas remove the word “Type of the Paper” form the manuscript title
- Please add BORSA infections to the list of keywords
- Line 27 : The abbreviation "CRP" should write in full , the same with PBP2a test in line 29
- Line 62 – 64 : please provide a supporting citation for mentioned statement
- Line 75: please kindly changes the word: Case presentation “to the case presentations and results
- Table 1: Please add a column for citations/references corresponding to each entry you refer to in the table.
- Adjust the column widths in Table 1 to improve readability
- Please include the ethical approvals and Consent information for this study
- Please move the Annexe 1 to the case presentations and results sections.
- The conclusion section is poor and does not effectively summarize the key findings of the study or its implications. Please rewrite the conclusion part in separate sections
- Please carefully check the references throughout the manuscript. Some references are missing critical information or are incomplete—for example, reference 10 (CLSI Performance Standards) lacks full publication details. Additionally, a few references appear outdated; please ensure the most recent and relevant literature is cited to support your statements
Author Response
This manuscript describes a case of BORSA. in a 39 old woman, analyse the genetic characteristics of the incriminated isolate and perform a retrospective analysis of BORSA strains submitted to the Belgian National Reference Centre for S. aureus. The manuscript is relevant, interesting, and addresses BORSA infections and their alternative therapeutic management. However, the manuscript requires improvement before consider for publication.
The authors would like to express their gratitude to the reviewer for dedicating time to review and provide feedback on our article. Your feedback has been carefully considered, and it has contributed to enhancing the quality of the article.
Comment for authors
- Line 2: pleas remove the word “Type of the Paper” form the manuscript title
It wasn't included in the original manuscript submitted and we've naturally removed it.
- Please add BORSA infections to the list of keywords
We have removed “borderline” and added “BORSA” following your comment.
- Line 27 : The abbreviation "CRP" should write in full , the same with PBP2a test in line 29
This has been adapted. Here is part of the paragraph concerned with the changes : « On examination, she was afebrile. Laboratory tests showed elevated C-reactive protein, and in-creased D-dimers. Later, she developed a septic peripheral venous thrombophlebitis, and treat-ment was adjusted to amoxicillin-clavulanate. Blood cultures grew S. aureus and antibiotic therapy was shifted to intravenous oxacillin following a negative Penicillin Binding Protein 2a test. »
- Line 62 – 64 : please provide a supporting citation for mentioned statement
Thank you for pointing that out. The reference was positioned one sentence later in the text but we've integrated it right after the sentence for greater clarity:
« Some authors have proposed that BORSA infections are generally more severe than those associated with Methicillin-sensitive S. aureus (MSSA) are, although this remains a topic of ongoing debate in the scientific community. (6) »
- Hryniewicz MM, Garbacz K. Borderline oxacillin-resistant Staphylococcus aureus (BORSA) – a more common problem than expected? Vol. 66, Journal of Medical Microbiology. Microbiology Society; 2017. p. 1367–73.
- Line 75: please kindly changes the word: Case presentation “to the case presentations and results
This has been adapted in the text.
- Table 1: Please add a column for citations/references corresponding to each entry you refer to in the table.
We have added a “references” column and referenced each paper associated with the data in the row.
- Adjust the column widths in Table 1 to improve readability
We've tried to adapt the table as best we could.
- Please include the ethical approvals and Consent information for this study
In accordance with Belgian legislation, ethical approval was not required, as the patient had provided informed consent and the article did not contain identifiable data and information on patient consent was already included in the text :
« Informed Consent Statement
Informed consent was obtained from the study subject. Written informed consent has been obtained from the patient to publish this paper. »
- Please move the Annexe 1 to the case presentations and results sections.
We have included the annexe in the body of the text in the “case presentation and results” section. We have then modified the table numbers in the captions and in the main text.
- The conclusion section is poor and does not effectively summarize the key findings of the study or its implications. Please rewrite the conclusion part in separate sections
Thank you for pointing that out. We've added a hopefully clear and precise conclusion in an extra paragraph:
« Conclusion
This case highlights the diagnostic and therapeutic challenges posed by BORSA, which is often misclassified by automated systems. Our investigation indicates that dis-crepancies between oxacillin and cefoxitin susceptibility may serve as a potential warning sign and should prompt further testing for BORSA. These findings underscore the need to integrate both phenotypic and molecular approaches for accurate detection and appropri-ate clinical management. Therapeutic strategies remain inconsistent, as standard treat-ment guidelines are lacking. Our findings support individualized management and call for further genomic and clinical research to define BORSA's resistance mechanisms and inform standardized diagnostic and therapeutic protocols. »
- Please carefully check the references throughout the manuscript. Some references are missing critical information or are incomplete—for example, reference 10 (CLSI Performance Standards) lacks full publication details. Additionally, a few references appear outdated; please ensure the most recent and relevant literature is cited to support your statements
Reference 10 has been updated. With regard to the dates of the references used, it should be noted that this is the case in recent articles on BORSA. Indeed, due to the lack of literature and investigations on the subject, we are often forced to use older sources, as there are very few of these available.
Reviewer 2 Report
Comments and Suggestions for Authors
This article summarizes the detection and treatment of very few cases of Borderline oxacillin-resistant Staphylococcus aureus infection reports in the literature and authors’ approach for one example. Due to the limited number of systematic studies on BORSA infections and the importance of the matter, the article will catch the attention of a wide range of researchers. However, there some minor issues that need to be resolved before the publication of this work:
- The keywords must be re-evaluated, “borderline” is not an appropriate keyword, “BORSA” is recommended to be added.
- Which studies of Table 1 correspond to NRC-1-3 strains of Table-2? More information on these strains, their isolation and case studies must be given.
- Table 2 also lacks proper controls. It does not lead to any control experiment to differentiate between BORSA and MRSA cases.
- The most important weakness of the manuscript is the lack of a clear treatment recommendation. Authors need to deepen the discussion of different treatment approaches and highlight the advantages and disadvantages or positives and negatives of each approach in a comparative manner.
Author Response
This article summarizes the detection and treatment of very few cases of Borderline oxacillin-resistant Staphylococcus aureus infection reports in the literature and authors’ approach for one example. Due to the limited number of systematic studies on BORSA infections and the importance of the matter, the article will catch the attention of a wide range of researchers. However, there some minor issues that need to be resolved before the publication of this work:
The authors would like to express their gratitude to the reviewer for dedicating time to review and provide feedback on our article. Your feedback has been carefully considered, and it has contributed to enhancing the quality of the article.
- The keywords must be re-evaluated, “borderline” is not an appropriate keyword, “BORSA” is recommended to be added.
We have removed “borderline” and added “BORSA” following your comment.
- Which studies of Table 1 correspond to NRC-1-3 strains of Table-2? More information on these strains, their isolation and case studies must be given.
The strains in the former “Table 2” (the numbers have been adapted following a reviewer's remark) are from the Belgian Staphyloccocus National Reference Center, with whom we collaborated for this article. The data relating to these strains were not published in the literature prior to the writing of our article. As these strains were used to find a way to detect them that could be used everywhere, we have not included the clinical information relating to them in the article. However, we have added some information about the CNR's missions to help readers understand them better :
« To this end, the Belgian NRC provided four confirmed BORSA isolates. The Belgian NRC for S. aureus is a center that voluntarily analyses and characterizes unusual phenotypes submitted by clinical laboratories across Belgium. As a significant number of laboratories routinely dispatch strains for investigation, these facilities have accumulated a substantial collection of strains displaying atypical phenotypes, including BORSA.»
- Table 2 also lacks proper controls. It does not lead to any control experiment to differentiate between BORSA and MRSA cases.
There is, in fact, no “control” as such for the detection of BORSA - which is the whole point of our article, highlighting just how difficult it can be to diagnose BORSA. All the necessary adjustments to exclude MRSA have been investigated at the CNR for all these strains.
- The most important weakness of the manuscript is the lack of a clear treatment recommendation. Authors need to deepen the discussion of different treatment approaches and highlight the advantages and disadvantages or positives and negatives of each approach in a comparative manner.
The literature on the subject is controversial, with few papers on the subject being published. Most of these papers highlight just how complicated it can be to manage antibiotic therapy for patients with this type of infection. The table 'Clinical and microbiological characteristics of various BORSA strains responsible for bacteraemia and/or endocarditis referenced in the literature' outlines the various treatments used and the outcomes of each case. It can therefore serve as a tool for the management of patients. We believe that future studies on this subject will help to characterise BORSA correctly and, hopefully, lead to a consensus on how to treat these infections. We've tried to be as complete and clear as possible on this subject, and hope you find it useful.
Reviewer 3 Report
Comments and Suggestions for Authors
General Comments
Buffart et al. report a borderline oxacillin-resistant Staphylococcus aureus (BORSA) clinical isolate from a patient with peripheral venous septic thrombophlebitis. BORSA strains are relatively rare and pose a significant challenge to clinical microbiology laboratories due to the absence of standardized criteria for their definition. Therefore, a case report including detailed phenotypic analysis and whole-genome sequencing data is of potential value.
I have no major criticisms, but I offer the following suggestions for improvement:
Subtitle ("review of literature"):
The subtitle includes "review of literature," which seems unnecessary in the context of this case report. While the discussion references previous reports, it reads as a standard case report discussion rather than a formal literature review. Please consider removing this phrase.
SNP detection – reference strain:
Please specify which strain was used as the reference for SNP detection.
Mutations in other genes:
While mutations in PBP are described, are there any mutations in other genes? Prior studies have reported mutations in gdpP (a phosphodiesterase that hydrolyzes the secondary messenger c-di-AMP) associated with BORSA.
Lines 171–176 (BORSA detection method):
Lines 171–172 state: "we conducted an investigation to ascertain whether BORSA could be detected using the tools already available in our laboratory." Please clarify the outcome of this validation. Does the content in lines 173–176 reflect the conclusion of that investigation?
NRC-confirmed BORSA strains 1–3:
The manuscript mentions NRC-confirmed BORSA strains 1–3, but their origin, related references, or public database accession numbers are not provided. Please include this information for transparency.
Figure 1:
I recommend adjusting Figure 1 so that the intervals between events more accurately reflect the actual passage of time, if feasible.
Table 1:
The sentence in lines 147–148 appears to refer to the origin of each isolate listed in Table 1. I recommend deleting this sentence and instead adding a dedicated column in Table 1 indicating the reference for each isolate. Additionally, I suggest moving this table to the supplementary material.
The manuscript would greatly benefit from improved clarity and precision in writing and presentation. I recommend a full review by a professional English editor or a native English speaker.
Author Response
Reviewer 3
General Comments
Buffart et al. report a borderline oxacillin-resistant Staphylococcus aureus (BORSA) clinical isolate from a patient with peripheral venous septic thrombophlebitis. BORSA strains are relatively rare and pose a significant challenge to clinical microbiology laboratories due to the absence of standardized criteria for their definition. Therefore, a case report including detailed phenotypic analysis and whole-genome sequencing data is of potential value.
I have no major criticisms, but I offer the following suggestions for improvement:
The authors would like to express their gratitude to the reviewer for dedicating time to review and provide feedback on our article. Your feedback has been carefully considered, and it has contributed to enhancing the quality of the article.
Subtitle ("review of literature"):
The subtitle includes "review of literature," which seems unnecessary in the context of this case report. While the discussion references previous reports, it reads as a standard case report discussion rather than a formal literature review. Please consider removing this phrase.
Thank you for your pertinent comment. We have removed it from the title.
SNP detection – reference strain:
Please specify which strain was used as the reference for SNP detection.
I'm not sure I fully understand that comment. We have not conducted any SNP detection. Please do not hesitate to let me know if I can clarify anything about this.
Mutations in other genes:
While mutations in PBP are described, are there any mutations in other genes? Prior studies have reported mutations in gdpP (a phosphodiesterase that hydrolyzes the secondary messenger c-di-AMP) associated with BORSA.
We used Resfinder for mutation analysis. Following your comment, we investigated whether we could identify any additional mutations. To date, none has been found.
Lines 171–176 (BORSA detection method):
Lines 171–172 state: "we conducted an investigation to ascertain whether BORSA could be detected using the tools already available in our laboratory." Please clarify the outcome of this validation. Does the content in lines 173–176 reflect the conclusion of that investigation?
These are the investigations we have carried out and which we describe in this article. There was no prior validation of this study.
NRC-confirmed BORSA strains 1–3:
The manuscript mentions NRC-confirmed BORSA strains 1–3, but their origin, related references, or public database accession numbers are not provided. Please include this information for transparency.
As these strains were used to find a way to detect them that could be used everywhere, we have not included the clinical information relating to them in the article. However, we have added some information about the CNR's missions to help readers understand them better :
« To this end, the Belgian NRC provided four confirmed BORSA isolates. The Belgian NRC for S. aureus is a center that voluntarily analyses and characterizes unusual phenotypes submitted by clinical laboratories across Belgium. As a significant number of laboratories routinely dispatch strains for investigation, these facilities have accumulated a substantial collection of strains displaying atypical phenotypes, including BORSA.»
Figure 1:
I recommend adjusting Figure 1 so that the intervals between events more accurately reflect the actual passage of time, if feasible.
The figure has been adapted by modifying the intervals to reflect the actual time between each event.
Table 1:
The sentence in lines 147–148 appears to refer to the origin of each isolate listed in Table 1. I recommend deleting this sentence and instead adding a dedicated column in Table 1 indicating the reference for each isolate. Additionally, I suggest moving this table to the supplementary material.
Following your pertinent comment, we have added a column indicating the article references associated with the various strains described in the table. However, it is this author's opinion that the aforementioned table is more suitably placed within the main body of the text, as it explicitly highlights a large number of details relating to the diagnosis and treatment of BORSA.
Comments on the Quality of English Language
The manuscript would greatly benefit from improved clarity and precision in writing and presentation. I recommend a full review by a professional English editor or a native English speaker.
Thank you for your feedback. Following this, a review of the quality of the English language was carried out. We hope that this will suit your requirements.